# Interpretation of Funerary Spaces in Roman Times: Insights from a Nucleus of Braga, NW Iberian Peninsula

Cristina Braga [1], Jorge Ribeiro [2,*] , Luis Fontes [2] and Ana Fragata [3,*]

1 ERA Arqueologia S.A., Rua Padre Cruz nº26, 4050-219 Porto, Portugal; cristinabraga@era-arqueologia.pt
2 Lab2PT/IN2PAST, Institute of Social Sciences, University of Minho, Campus de Gualtar, 4710-057 Braga, Portugal; lfontes@uaum.uminho.pt
3 GeoBioTec, Geosciences Department, University of Aveiro, Campus de Santiago, 3810-193 Aveiro, Portugal
* Correspondence: jribeiro@uaum.uminho.pt (J.R.); afragata@ua.pt (A.F.)

**Abstract:** The funerary/cult archeological nucleus of Rua do Raio (Braga, in the northwest of the Iberian Peninsula) was discovered between 2007 and 2009, under the excavation works of a necropolis of *Bracara Augusta*. This building exhibits a set of particularities that confirms its archeological importance. It is a construction dating from the middle of the 1st century AD, subject to two reforms, one in the second half of the 1st century AD and another in the 2nd century AD, with a trapezoidal shape and comprising ten rectangular tanks rendered with *opus signinum* mortars. It shows a unique architectural configuration in the city, as well as in the Portuguese territory, and the space is under musealization, together with a set of graves identified in the same archeological intervention. The present investigation contextualizes the funerary and religious architecture of the city. Its description is presented, highlighting its relevance, and an interpretative possibility is formulated.

**Keywords:** archeology; Roman times; Hispania; *Bracara Augusta*; funerary buildings; funerary practices

## 1. Introduction

The archeological nucleus of "Rua do Raio" was identified during an archeological intervention carried out in Braga, between 2007 and 2009, in the Post Office block (CTT) denominated "N.A.2", directed by the Archeology Unit of the University of Minho (UAUM), under the urban rehabilitation of the CTT building (Martins et al. 2010). The work carried out allowed the exhumation of several notable archeological remains, including a necropolis area associated with a road that connected *Bracara Augusta* to *Asturica Augusta* and with a glass workshop. The N.A.2 (Building R05) occupies an area of roughly 200 m², in the southeastern part of this CTT block. The collected data point to its foundation in the beginning of the 1st century AD, in the northeast of the ancient Roman city, followed by two reforms in the second half of the 1st century AD and in the 2nd century AD. The building was abandoned in the 3rd century AD.

This funerary/cult building is a remarkable construction, without any parallel in the city, is part of the city's oldest archeological remains. Its scientific and patrimonial value led to an in situ preservation and later to a musealization process and integration in the new construction, which is still in progress.

Its function is still unknown; however, the "Idolo Fountain" (*Fonte do Ídolo* in Portuguese) (Garrido Elena et al. 2008; Tranoy 1981) in its vicinity, which is an important Romanized indigenous sanctuary, and its enclosure in a necropolis area, suggests a connection to funerary or cultic practices.

This work is structured in three parts: in the first part, an archeological framework of the subject under analysis is carried out, with a focus on the components of funerary and religious architecture; the second part emphasizes the built sequence transformation of

the building under analysis (N.A.2 or Building R05); the third part focuses on the N.A.2 construction process, architecture, materials, function, and musealization process.

## 2. Archeological Setting: *Bracara Augusta*, Necropolises, Religious Buildings, and Sanctuaries

The city of *Bracara Augusta* corresponds to a planned urban space created by the emperor Augustus in the northwest of the Iberian Peninsula (Figure 1), after the Cantabrian Wars at the end of the 1st century BC (Le Roux 1994; de Sande Lemos 2002; Martins and Carvalho 2017; Rodríguez Colmenero 1996). The characteristics of the city have been revealed by dozens of excavations carried out since the 1970s, under the *Bracara Augusta* Project (Martins et al. 2020). Thus, the works carried out and published reveal that it was an important administrative, economic and political center in the Roman period, being the capital of *conventus* in the high empire period and, later, at the end of the 3rd century, becoming the capital of the new province of *Gallaecia*, under the domain of the Emperor Diocletian (Martins and Carvalho 2010; Martins et al. 2020; Morais 2009, 2010).

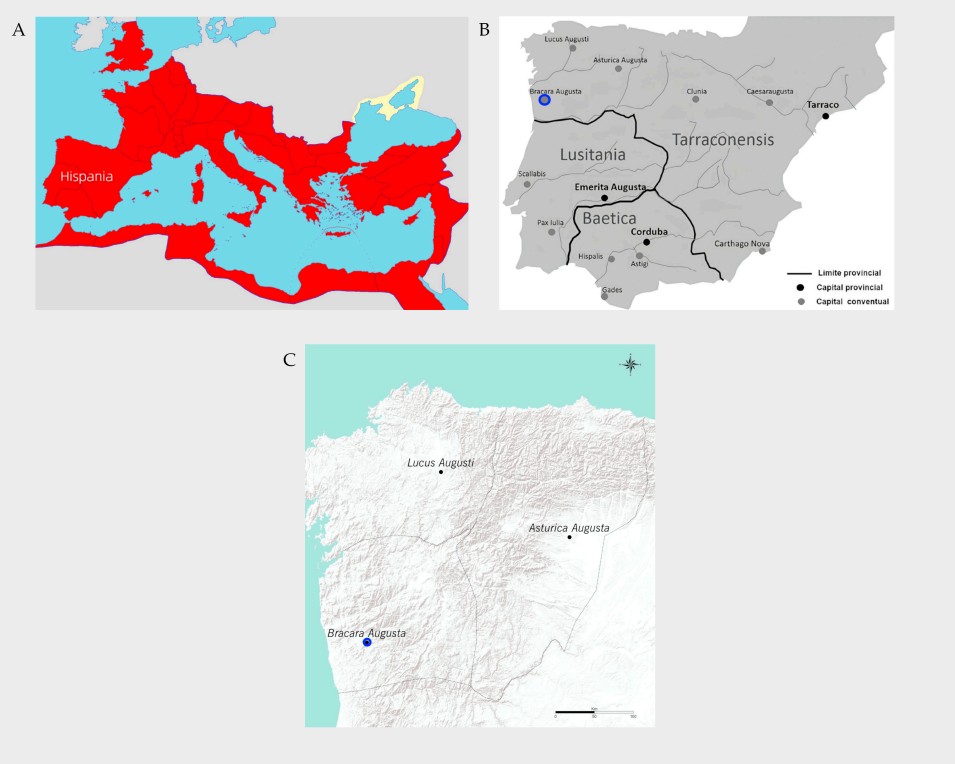

**Figure 1.** (**A**) Location of Hispania in the Roman world and (**B**,**C**) *Bracara Augusta* in Hispania and in the northwest of the Iberian Peninsula (Braga 2018).

In the city of Braga, there are several necropolis spaces (Figure 2). As usual, in Roman cities, these spaces were located outside the walls of the cities, near the main or the secondary roads. In the city, there is the Maximinos necropolis (Via XX and XVI), the Rodovia necropolis (*Via Bracara Augusta—Emerita Augusta*), the Campo da Vinha *nuclei* (Via XIX/XVIII), and the necropolis of Via XVII (Via *Bracara Augusta—Asturica Augusta*), where the intervention area of the former Post Office block (CTT), under the present investigation is located, including the archeological nucleus of Dr. Gonçalo Sampaio street and the nucleus of Cangosta da Palha (Braga and Martins 2015; Vaz et al. 2021, and references therein) (Figure 1). The investigation of Braga's Roman necropolis has been recorded over the last nearly 40 years of archeological excavations in the city, under the *Bracara Augusta* Project (Martins 2014; Martins and Carvalho 2017; do Carmo Ribeiro 2008) from the Archeology Unit of the University of Minho (UAUM). These excavations led to several

scientific works. One (Martins and Delgado 1989–1990) was dedicated to the funerary rituals, and the others investigated a necropolis nucleus of Via XVII, the city's funerary rituals and sepulchral spaces, and the rituals, uses, and funerary landscapes of the Via XVII necropolis (Braga 2018); and the ceramics of this nucleus (Morais et al. 2013b) from the 1st century AD, and the funerary topography of the Via XVII necropolis in Late Antiquity and the rituals and funerary spaces of *Bracara Augusta* (Braga and Martins 2015).

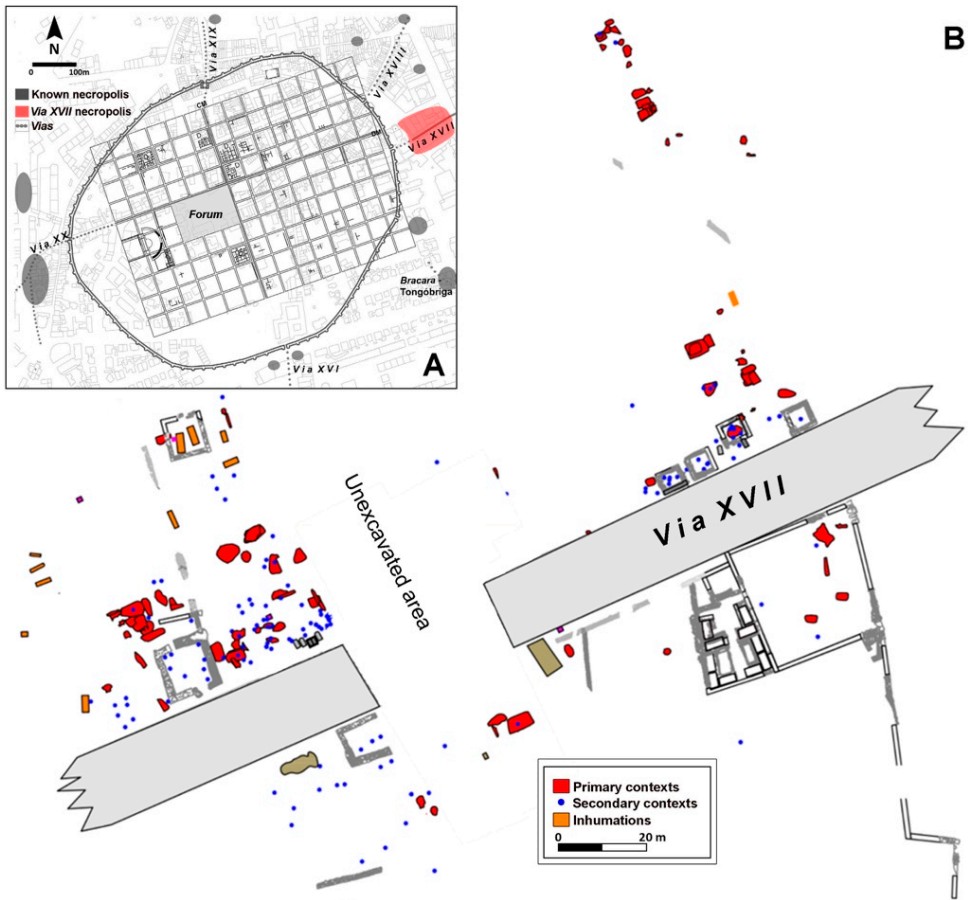

**Figure 2.** (**A**) Location of the necropolises and the excavation in the plan of *Bracara Augusta* and (**B**) detailed plan of the Via XVII necropolis excavation (Adapted from (Vaz et al. 2021)).

As in other Roman cities, the built space of *Bracara Augusta* shows several religious buildings and sanctuaries, although in most cases only partial remains were found. The *Fonte do Ídolo* stands out, monumentalized in Roman times, and dedicated to *Nabia*, a deity associated with water, fertility, and nature (Garrido Elena et al. 2008; Martins et al. 2012a). Another sanctuary, possibly a *fanum*, composed of a cylindrical column-shaped altar and engraved with the word *sacrum*, was discovered in the peripheric area of the city, in the context of construction, together with clay building materials (Carvalho et al. 2006; Morais 2010). It is also important to mention the city *forum*, indicated on the map of Braunio (Morais 2010; Morais et al. 2013a; do Carmo Ribeiro 2008) from the 16th century and linked to a set of monumental architectural elements (Martins et al. 2012a; Ribeiro 2018), with some of those elements assumed to belong to the religious buildings. Under Braga's Cathedral, another building was identified with a religious function identified by an inscription dedicated to *Isis Augusta* by a priestess of the imperial cult, *Lucrecia Fida*, which may reveal the existence of a temple dedicated to the oriental divinity (Martins et al. 2012a; Morais 2009–2010). In addition to the elements, monuments dedicated to the imperial cult are also documented, namely to Augustus, including a set of three pedestals of statues, of which only one was preserved (Martins et al. 2012a).

The *Fonte do Ídolo* ([Martins et al. 2012b](#)), due to the connection that it seems to have with the site under investigation, deserves special attention. It is a rock sanctuary located northeast of *Bracara Augusta*, which represents a monumentalized version of a *locus sacer* where water worship took place before Roman occupation. This monument dedicated to the indigenous goddess *Nabia* is carved with inscriptions and high-relief figurines. In the 1st century AD, the sanctuary was converted into a Roman monument, probably built by *Celicus Fronto*, whose name is engraved on the rock outcrop. In Flavian times this space was refurbished by descendants of *Celicus Fronto*. In the surrounding area of this sanctuary, Roman plumbing and reservoirs possibly fed by the fountain were found. There are several theories related to *Fonte do Ídolo*, although all agree that this space was a sanctuary dedicated to water divinity.

## 3. The Main Phases of the Necropolis Occupation

The occupation of this necropolis under analysis, in the ancient world, was between the Bronze Age and the Suevo-Visigothic period. Indeed, the first occupation is documented by a tomb from the Bronze Age. Then, in the period before the reign of Augustus, the several ditches opened in the rock substrate and the existence of a dirt road were eventually related to a previous plan of the construction of Via XVII, built in the beginning of the 1st century AD. In the same period, and associated with it, the necropolis was implemented. In the middle of the 1st century AD, N.A.2 (Building R05, Figure [3](#)) was built and subject to a first remodeling in the second half of the same century, with its internal compartmentalization. During the following century, the building benefited from a second reform, where a new compartmentalized space was built. During the second half of the 3rd century AD and the beginning of the 4th century AD, this necropolis space was less used, probably due to the existence of other burial areas in the city. The abandonment of N.A.2, as a funerary space, was in the 4th or 5th centuries AD, when a glass workshop was built on top of it. Between the 5th and 7th centuries AD, this entire area was no longer used as a necropolis.

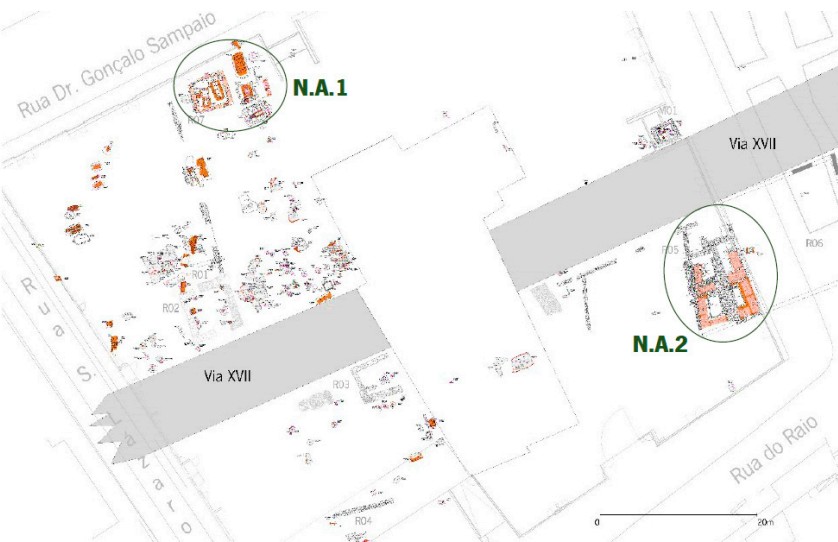

**Figure 3.** Location of N.A.2 (Building R05).

## 4. The Constructive Process of the N.A.2 in the Necropolis

*4.1. The Main Phases of Occupation*

The N.A.2 was built in the middle of the 1st century AD (Phase I), having benefited from two reforms (Phases II and III); at the end of the 2nd century AD it was abandoned ([Braga 2018](#)).

### 4.1.1. Phase I

As already noted, the N.A.2 began to be built in the middle of the 1st century AD. At this stage (Phase I) and from a constructive point of view, only the outer walls constructed from granite stone, with a poorly carved shape, and the interstices filled with sandy soil and fine grave, were identified. The base of the perimeter wall made of small-sized granite stone was also recognized (although with some constructive lacunae), next to the west limit of the building (Figure 4A).

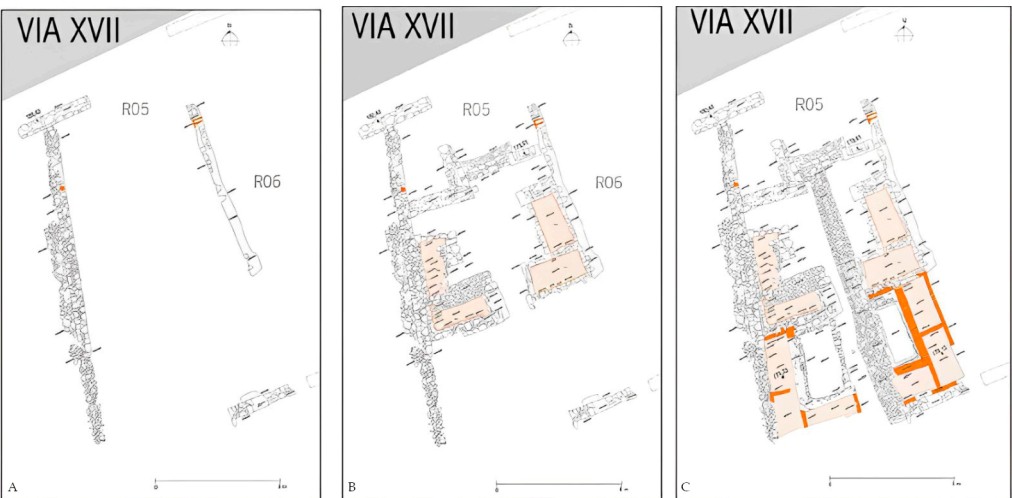

**Figure 4.** Evolution of N.A.2 reforms throughout (**A**) Phase I; (**B**) Phase II; (**C**) Phase III.

### 4.1.2. Phase II

In the second half of the 1st century AD (Phase II), the building experienced the first internal compartmentation via the construction of four compartments made of granite stone masonry with a rectangular shape and coated with *opus signinum* mortar (Figure 4B).

### 4.1.3. Phase III

The documented new reform of the N.A.2 in the 2nd century AD (Phase III) refers to the reformulation of the space, which was previously open, located to the south of the building. In this phase, six more compartments were constructed, replicating the model of the previous phase but with some constructive differences, and particularly the use of clay materials (bricks) for the execution of the walls. On both sides of the north–south dividing wall, a perfect symmetry was observed (Figure 4C).

### 4.2. Description of the Building (Architecture and Materials)

The archeological works documented the existence of a workshop associated with the glassmaking activity, where the walls were removed revealing the existence of a previous construction. Several compartments coated with *opus signinum* were recognized, ordered according to a west wall that was about 13 m long and oriented north/northwest.

The extension of the excavation works to the east limit made it possible to identify new compartments similar to those initially recovered, showing the presence of a cohesive building, with internal compartmentalization revealing a relative symmetry limited by two walls (east and west), with a wall at its northern limit oriented east/west and parallel to Via XVII (Figure 4C).

The N.A.2 is located on the south part of this necropolis area, on the south side of the road, and was built in close connection with the necropolis, from where the access to the N.A.2 takes place. On the opposite side of the road, there is a set of mausoleums.

Regarding its shape, the N.A.2 has an asymmetrical trapezoidal configuration with its internal structure organized around a central axis, and materialized through a wall oriented north-northeast/south-southwest. This is only perceptible by the existence of foundation

of the wall, which separates two areas with perfectly symmetrical compartments (Figure 5). These compartments configure a kind of unit having similar dimensions and constructive solutions, although with different orientations. These are ten watertight units coated with *opus signinum* mortars, lowered to the circulation quota, located at 179.91 m.

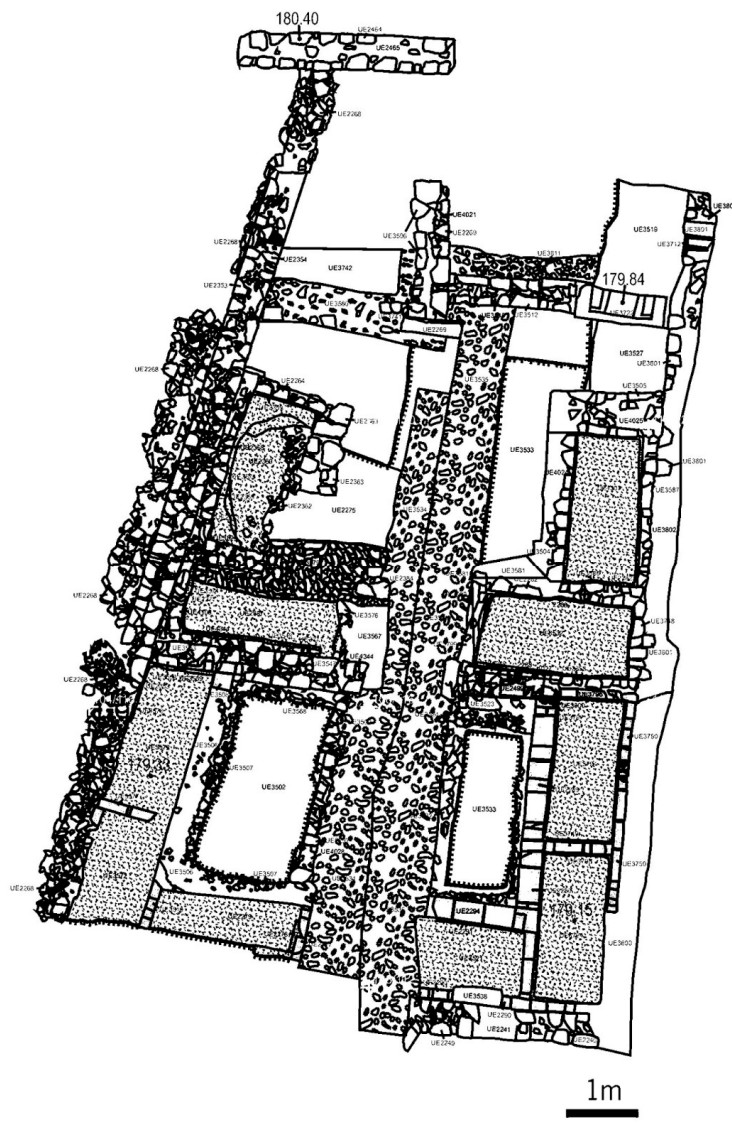

**Figure 5.** Plan of the trapezoidal N.A.2 (Building R05) under musealization.

The N.A.2 has a maximum length of 14.22 m, and a width of 8.62 m on the south side and around 6 m on the north side. The compartments have very similar dimensions, varying between 2.10 × 0.90 m and 2.05 × 1.00 m, and were observed to be differently preserved in terms of height. The best-preserved compartment, located on the east side of the building, is about 0.80 m high. The walls and the pavement of all the compartments are coated with *opus signinum*.

The interior of the building can be divided into three distinct spaces, which we call the northern sector, the central sector, and the southern sector.

The access to the building was from the road, through the northern sector of the building. It is limited by the large wall that marked the south side of the road and presents a particular organization, composed of two contiguous compartments with different dimensions, which complete the main body of the building. In this area, a granite threshold was exhumed, confirming the circulation level inside the building, estimated to be at 179.90 m.

The central sector of the building is tight and made up of four compartments, configuring very homogeneous spaces, of about 2/2.15 m in length and 0.96/1.05 m in width, with two compartments on each side, forming an L due to its arrangement. Those compartments abutting to the lateral walls have their major axis oriented to the north/northwest, with the east/west-oriented ones located to the south. In this sector, there is also an open central space, around which the compartments are organized. These compartments show different conservation state and different construction systems. Those located on the east side are better preserved, and the one located on the north side has a maximum wall height of about 0.80 m and is completely covered with well-preserved *opus signinum* mortars. The heights of the base of the compartments, whose lowest quota is at 179.15 m, are 0.76 m below the level of the entrance threshold, which indicates that they were below the circulation level. Regarding the building materials, this sector does not use bricks in internal walls but rather irregular granite stone masonry. At the base, the *opus signinum* coating mortars cover a preparatory level formed by material comprising small stones.

The space located in the south of the building seems to be structured around a central open area. This space has six compartments, three on the east side and three on the west side, organized in an L shape. In fact, on both sides there is a succession of two contiguous spaces, with the major axis (2.10 m) oriented north/northeast, leaning against the lateral walls. Two others, lined up by the south wall of the building and oriented east/west, complete the set and define a narrow circulation corridor of around 1.12 m. In this space, the divisions of the compartments were executed with brick masonry, and the internal walls were made with the same material (Figure 6A). *Lydion* bricks were used with layers of yellowish mortar. The conservation state is variable, with the eastern receptacles have higher walls, something that is related to the construction of the craft space over the N.A.2, whose constructive impact focused on the west area. In any case, the quality of the *opus signinum* mortars that covers the bottom of the compartments is technically very good, showing a complex preparation that is still well preserved (Figure 6B).

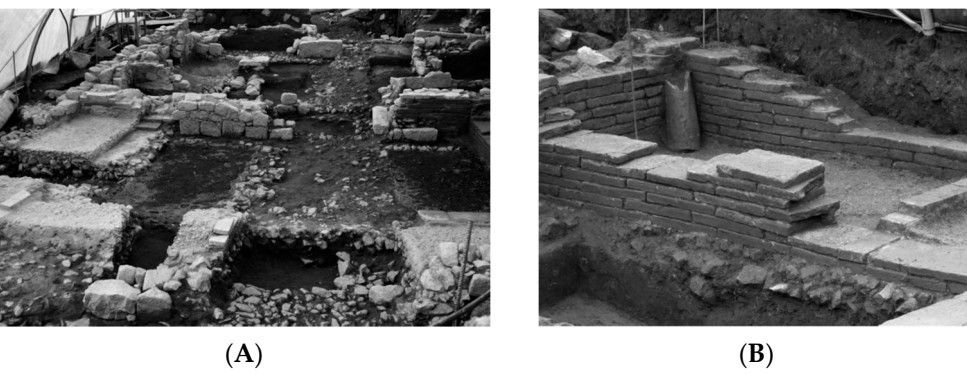

|   |   |
|---|---|
| (**A**) | (**B**) |

**Figure 6.** South view of the N.A.2 (**A**) at the end of the archeological excavation; (**B**) perspective of the compartments on the east side of the building.

As can be seen from the restored floor plan of the building and the description above, its organization, despite the asymmetry of its east and west walls, sets up an interior space designed and structured in a symmetrical way. On the other hand, the building seems to be part of a large enclosure, limited on the north side by the wall delimiting the road; this is blocked by another structure having a northwest orientation, identified next to the CTT block. The area between the building and the wall seems to configure a large open space, without other buildings, and with an undetermined function.

*4.3. The Function of N.A.2*

The data available for the interpretation of the N.A.2 consist of elements resulting from the archeological excavation process and do not allow the attribution of a function. A possible hypothesis could be its use as a craft space; however, in fact, none of the

compartments showed any element that suggests the existence of drainage or water entry, or even provided residual traces related to a possible use of this nature. Other hypotheses could be assumed, considering the structure of the building, its proximity to the *Fonte do Ídolo*, and the adjacent funerary structures, which suggest the possible association with funerary or cultic practices although, until now, with an undefined nature. It is pertinent to remember that this large area began to be structured at a time coincident with the first monumentalization project of *Fonte do Ídolo*, which is located 63 m south of Via XVII. It is possible to assume, therefore, that the emergence of the first forms of appropriation of the funerary space may have been triggered together with the monumentalization program of the sanctuary (Garrido Elena et al. 2008). The importance attributed to the water in this area of the city seems to have also been relevant to other structures possibly related to it, such as the archeological remains identified in the "Granjinhos area" located nearby, which included a heated tank and a heating system for channels dug in the bedrock (Martins and Ribeiro 2012). Certainly, there are no structures parallel to this funerary structure known in Braga.

Thus, we do not exclude the hypothesis that the building could be related to performance tributes and commemoration rituals related to the practice of cremation. There is a difficulty in finding parallels (e.g., similar structures) and the clear attribution of the structure's function. The existing parallel is only from a constructive point of view.

Despite the difficulties in ascribing a clear function to the building, it is important that some parallels are documented, although with different chronology and construction typology. One of the parallel cases is the necropolises of Cadiz, where structures with rectangular morphologies, covered with *opus signinum* mortars, were exhumed. Here, these structures, defined as "swimming pools", are located outside the walls in the middle of the funerary space, and are mostly built in an isolated form, and in some cases associated with conduits and wells. They feature a staggered access system, located at the ends of the box, with dimensions that reach a length of 4 m by a width of 1 m. These structures date from a precise chronological period between the 2nd and 1st centuries BC, and disappeared in the imperial era. These spaces are linked to purification cults of individuals after funeral ceremonies, and this typology of buildings is related to Cadiz tradition and associated with lustral rituals, strongly influenced by the funerary patterns of Oriental and Late-Punic nature (Villedary y Mariñas and Gómez Fernández 2010).

### 4.4. The Musealization Project of the N.A.2

The characteristics of the N.A.2 enclosure, and its uniqueness, state of preservation, and construction chronology, which bring it closer to the moment of the foundation of *Bracara Augusta*, led the scientific direction of the archeological works to assume the importance of its conservation in situ, for later musealization, as it constitutes an architectural construction with unique features.

The enclosure was properly protected to proceed with the excavation works for the ongoing musealization project. With the redefinition of the architectural design of the shopping center, it was possible to delimit and integrate the N.A.2 site into a closed and protected space (Figure 7).

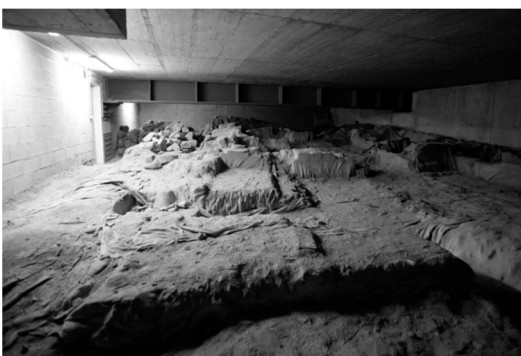

**Figure 7.** The perspective of the N.A.2 (Building R05) ruins.

## 5. Conclusions

The N.A.2 was built in the middle of the 1st century AD in one of the largest necropolises of *Bracara Augusta*, which was one of the conventual capitals of the province of *Tarraconensis*. It was discovered under a major archeological intervention work carried out between 2007 and 2009 in the city of Braga. The enclosure was remodeled twice, in the second half of the 1st century AD and in the 2nd century AD. The trapezoidal building is located immediately to the southeast of the Roman road that leads to *Asturica Augusta*, and has an NW–SE alignment. The building's final plan is structured around a dividing wall, configuring symmetrical spaces in which five watertight tanks with *opus signinum* coating were inserted on each side of it, creating a total of ten compartments. In the 3rd century AD, the N.A.2 was abandoned, and an artisan space for glass production was built.

Its unique character, its location in a necropolis area, and its construction near *Fonte do Ídolo* (an indigenous sanctuary of worship and adoration) indicate a possible funerary use or cult that needs to be investigated. The uniqueness of N.A.2, its state of conservation, and the chronology of its construction in the first decades of the life of the Roman city motivated its in situ preservation and the definition of a musealization project, which is still in progress, to make it more accessible and visitable.

The in-depth study of N.A.2, in parallel with the analysis of other similar spaces identified elsewhere in the Iberian Peninsula, still needs to be conducted to refine this first investigation, especially regarding the function of this space.

Concerning the constructive and material aspects of this building, studies are currently ongoing regarding the chemical and mineralogical characterization of the mortars using X-ray diffraction (XRD), X-ray fluorescence (XRF), and scanning electron microscopy (SEM) of the walls, as well as the coatings made with *opus signinum*. These studies are being performed by a multidisciplinary team from the University of Minho and the University of Aveiro, and show some similar aspects to other preserved archeological sets (N.A.1) in *Bracara Augusta*, a burial area of the Via XVII necropolis, which consists of five burial graves, and a rectangular enclosure (Figure 3), and is also in the process of musealization (Fragata et al. 2021; Ribeiro 2013).

**Author Contributions:** Conceptualization, C.B., J.R., L.F. and A.F.; methodology, C.B., L.F., J.R. and A.F.; validation, C.B., J.R. and A.F.; formal analyses, J.R. and A.F.; investigation, C.B., J.R., L.F. and A.F.; writing—original draft preparation, C.B., J.R., L.F. and A.F.; writing—review and editing, C.B., J.R. and A.F.; supervision, C.B., J.R. and A.F. All authors have read and agreed to the published version of the manuscript.

**Funding:** This research was supported by the Landscape, Heritage and Territory Laboratory (Lab2PT), Ref. UIDB/04509/2020, financed by national funds (PIDDAC) through the FCT/MCTES, and the Geo-BioSciences, GeoTechnologies and GeoEngineering Research Centre (GeoBioTec), Ref. UIDB/04035/2020, funded by FCT and FEDER funds through the Operational Program Competitiveness Factors COMPETE and by national funds (OE), through FCT in the scope of the framework contract foreseen in the numbers 4, 5 and 6 of the article 23, of the Decree-Law 57/2016, of August 29, changed by Law 57/2017, of July 19.

**Institutional Review Board Statement:** Not applicable.

**Informed Consent Statement:** Not applicable.

**Data Availability Statement:** Not applicable.

**Acknowledgments:** The authors are grateful to Liberdade Street Fashion Shopping Centre—Cuman and Wakefield, and especially to its manager José Alberto Martins, for facilitating the access to the funerary nucleus, to the Archaeology Unit of University of Minho (UAUM), for the technical assistance.

**Conflicts of Interest:** The authors declare no conflict of interest.

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
