# Peer review of "Interpretation of Funerary Spaces in Roman Times: Insights from a Nucleus of Braga, NW Iberian Peninsula"

_religions, doi:10.3390/rel14091185_

Round 1
Reviewer 1 Report
The study is very original and comes to fill a gap in the investigations on the city of Braga.
Author Response
Response to Reviewer 1 Comments:
The authors are very thankful for the revision. The manuscript was revised according to the reviewers’ comments, using the uploaded version from the link that you have indicated, and the English and content revisions were marked up using the “Track Changes”.
Reviewer 2 Report
A:) Maps and plans:
The study would be improved if it could contain:
- the location of the city on the peninsula
- the location of the excavation in the urban plan of the Roman city and the location of the necropolises
B:) Suggestions for improvement:
- Regarding the date of the foundation of the city should mention studies prior to Lemos 2002, because as it is, it gives the impression that this author was the first....
- In the introduction, 2nd paragraph, the type of building identified is missing;
- In point 2, you mention the economic importance of the city. Here you should refer to a synthesis study:
MORAIS, R. (2005). Autarcia e Comércio em Bracara Augusta. Contributo para o estudo económico da cidade no período Alto-Imperial, Bracara Augusta, Escavações Arqueológicas 2, Braga: UAUM/Narq, Volume I (470 p.), Volume II (487 p.) ISBN 972928214
C: Missing bibliography (in addition to the above):
MORAIS, R. (2010). Bracara Augusta. Braga: Edição da Câmara Municipal de Braga (205 p.). Depósito Legal: 311722/10
MORAIS, R., FERNÁNDEZ FERNÁNDEZ, A. & BRAGA, C. (2013). Contextos cerámicos de la transición de Era y de la primera mitad del s. I provenientes de la necrópolis de la Vía XVII de Bracara Augusta (Braga, Portugal). Congrès International de D'Amiens – SFECAG (9-12 maio). Amiens, pp. 313-326. ISBN 555-2-00-185719-0
Author Response
Response to Reviewer 2 Comments:
The authors are very thankful for the revision. The manuscript was revised according to the reviewers’ comments, using the uploaded version from the link that you have indicated, and the English and content revisions were marked up using the “Track Changes”. Please find attached the content included to answer to the recommendations of reviewer 2.

Reviewer 3 Report
The case study is quite interesting and indeed it has potential considering the presence of an important cultic complex in the same area. The paper’s main contribution is the presentation of a previously unknown building which has some rather unusual, yet interesting, architectural features. Although immediate parallels to its structure do not exist, the identification of a similar building in Cadiz (Andalucia, Spain) seems to provide an interesting comparison, worth further investigation, which may broaden the scope of the presented research. Also interesting is the afterlife of the building and its conversion into a glass workshop in the late Antiquity following its abandonment in the 3rd century AD.
The argument is very speculative at this stage. The evidence presented, possibly because still partial, impairs the testability of the hypothesis put forward. The discussion is not always presented in a well-structured manner, and this prejudices a full appreciation of the case study and of its results. While the architectural features are described in-depth, the argument lacks any theoretical foundation in support of what is claimed in the title. The word ‘materiality’ is used but there is not engagement with the theoretical debate, nor is a real interpretation of this and other funerary spaces provided in support of the building’s supposed ‘religious’ or ‘funerary’ meaning. The conclusions are overall consistent with the evidence and arguments presented, but introduce an N.A.1 monument, whose plan is provided without explaining what it is.
Major revision of the structure, contents, methodology and language are therefore strongly recommended. A list of amendments is provided in the attached report.

A thorough revision of the English grammar, syntax and lexicon is essential.
The choice of certain English terms to translate non-English ones often makes the meaning unclear.
Author Response
Response to Reviewer 3 Comments:
The authors are very thankful for the revision. The manuscript was revised according to the reviewers’ comments, using the uploaded version from the link that you have indicated, and the English and content revisions were marked up using the “Track Changes”. Please find attached the content included to answer to the recommendations of reviewer 3.

Round 2
Reviewer 3 Report
The reviewer appreciates the authors' efforts to make the suggested changes. The presentation of the case study is much clearer and its significance is now better understood.
Despite the obvious attempts at correcting the language, the quality of English is still very low, and this impairs full comprehension of some parts of the text. A thorough revision by a native speaker is strongly advised.
Author Response
The authors are very thankful to reviewer 3 for the revision.
The manuscript was revised according to the reviewers’ comments, using the uploaded version from the link.
The recomendations from reviewer 3 were included in the revised manuscript using track changes. The manuscript was subject to major English revision.